# Intermediate Temperature PEFC’s with Nafion^®^ 211 Membrane Electrolytes: An Experimental and Numerical Study

**DOI:** 10.3390/membranes12040430

**Published:** 2022-04-15

**Authors:** Oliver Fernihough, Mohammed S. Ismail, Ahmad El-kharouf

**Affiliations:** 1School of Chemical Engineering, University of Birmingham, Edgbaston, Birmingham B15 2TT, UK; a.el-kharouf@bham.ac.uk; 2Energy Institute, University of Sheffield, Sheffield S3 7RD, UK; m.s.ismail@sheffield.ac.uk

**Keywords:** polymer electrolyte fuel cells, proton exchange membrane, fuel cell modelling, Nafion, intermediate temperature

## Abstract

This paper evaluates the performance of Nafion 211 at elevated temperatures up to 120 °C using an experimentally validated model. Increasing the fuel cell operating temperature could have many key benefits at the cell and system levels. However, current research excludes this due to issues with membrane durability. Modelling is used to investigate complex systems to gain further information that is challenging to obtain experimentally. Nafion 211 is shown to have some interesting characteristics at elevated temperatures previously unreported, the first of which is that the highest performance reported is at 100 °C and 100% relative humidity. The model was trained on the experimental data and then used to predict the behaviour in the membrane region to understand how the fuel cell performs at varying temperatures and pressures. The model showed that the best membrane performance comes from a 100 °C operating temperature, with much better performance yielded from a higher pressure of 3 bar.

## 1. Introduction

Increasing the polymer electrolyte fuel cells (PEFCs) operating temperature is an attractive proposition. The most significant portion of the research effort has been focused on cathode catalyst material and layer optimisation. The cathode kinetics are four times slower than the anode [1] and are therefore considered the cell reaction rate-determining step [2,3,4]. Therefore, it is a crucial area of research. The majority of research is focused on conventional operating temperatures (60–80 °C); however, increasing the operating temperature also improves cathode kinetics, as reported by Song et al. [5].

Moreover, higher temperatures increase the saturation of water vapour and decrease the likelihood of water flooding. The elevated temperatures also improve the fuel cell tolerance to CO impurities in hydrogen; increased temperature makes the adsorption of CO to the platinum surface unfavourable [6,7,8]. As with many transport phenomena, oxygen transport increases with temperature. The diffusion of oxygen in water is greater with higher temperatures [9], enabling the transfer of reactants to the catalyst active sites to be significantly boosted, which has a significant knock-on effect on performance.

Perflurosulphonic acid (PFSA)-based membranes, namely, Nafion, are today’s PEFC industry standard [10]. The central role of the membrane is to separate electronic and ionic charges and provide an impermeable barrier to the gases of the anode and cathode [11]. PFSA membranes are made of a hydrophobic polytetrofluoroethylene (PTFE) backbone and a side chain terminating in a sulphonic acid group, which requires high hydration to conduct protons [12]. The structure has been described with a pore and chamber model by Mologin D et al. [13].

Therefore, the membrane is the main component limiting the operational temperature of PEFCs. Higher temperatures mean less dissolved water and more steam. This phase difference significantly affects water transport through the cell, although it improves mass transport at the interface. The extra water movement reduces the membrane’s ability to accumulate and retain water and results in dehydration [14]. This has a detrimental effect on PEFC performance as the conductivity drops with reduced water content.

For low-temperature fuel cells, degradation of the membrane occurs through the following three primary mechanisms: chemical, thermal, and mechanical. Thermal degradation is the breakdown of the polymer chain from heat, such as thermal relaxation to the point of plastic deformation and pinhole formation at hot spots; mechanical degradation results from the swelling and contraction of the membrane with varying hydration levels and the pressure differential across the membrane [15]. Chemical degradation comes from weak sulphonic acid side chains that are vulnerable to peroxide attack. Peroxide is formed due to hydrogen crossover across the membrane, exacerbated by non-steady-state operation and time spent at open circuit voltage (OCV) [16]. Higher operating temperatures can enhance all membrane degradation mechanisms, reducing the cell’s lifetime. Therefore, researchers have been investigating alternatives or modifications to the currently used PFSAs [17,18,19,20].

Glass transition temperatures in polymers represent a temperature above which bulk morphological relaxation takes place. A relaxation of the polymer chains causes a destabilisation of the structure of the polymer. Small relaxations are elastic, but large relaxations can mean the complete separation of the crosslinking within the polymer structure. Nafion comprises the following two polymer chains: a backbone of PTFE and a side chain containing the sulphonic groups. The backbone has a glass transition temperature of 110–130 °C [21,22]. As discussed by Calleja et al. [21], the relaxation of the backbone has stages, and the relative change in relaxation is low until 125 °C. Sidechain relaxation is characterised by near temperatures of 100 °C [22], posing a problem for the performance of the membrane at temperatures above 100 °C because relaxation increases the porosity and decreases the stability of the membrane.

PSFA membrane development has focused on increasing the performance of the membrane while maintaining the desired lifetime for PEFC applications. Membrane thicknesses were reduced from Nafion 117, which had a thickness of 183 μm, to Nafion 211, at a thickness of 25 μm [23]. Improvements in manufacturing, such as the change from extrusion to dispersion casting, have yielded high-performance membranes with very high conductivities [24] whilst maintaining mechanical strength and durability. Reinforcement with other polymer fibres such as PTFE and PBI can increase mechanical strength and durability [25,26,27]. Finally, dopants such as cerium oxide are now used as radical scavengers to reduce the chemical degradation of the membrane [28], helping to extend its lifetime. The next area for development should be increasing the operating temperature; the benefits are abound [29].

Modelling is advantageous because it can reduce costs associated with designing and building prototypes and allow investigations beyond the limits of the current experiments. They can give insights into complex systems where experimentation can be difficult or expensive. The Springer et al. model [30] is the most famous empirical model for Nafion membranes; it enables one to estimate the ionic conductivity of the membrane electrolyte with the knowledge of water content and temperature. Many authors have used it as the foundation of membrane modelling [14,31,32,33]. The model developed by Springer et al. [34]. was partly based on Yeo and Eisenburg’s [35] measurements that described the activation energy and diffusion of water in Nafion 117.

The main limitation of the Springer et al. [30] model is that it is based on a limited range of temperatures (30–80 °C) and pressures (1 atm). However, many real-life fuel cells operate at higher pressures to increase their performance [36]. The behaviour of water extensively was studied, leading to the production of steam tables, which they show the thermodynamic properties of saturated steam [37]. Steam tables are helpful as they give insight into the relative behaviour of water at different pressures, temperatures, and saturation levels. The data surrounding water sorption into the Nafion phase at elevated temperatures and pressures are still lacking due to the complexity of the measurement experimentally.

Nafion 211 and Nafion 117 are similar as they share the same base polymer unit; however, they differ in thickness and production method as follows: Nafion 211 is 25 μm thick and made by dispersion casting, while Nafion 117 is 183 μm thick and made by the extrusion process. Dispersion casting results in a more relaxed polymer structure, which changes the water sorption and membrane swelling behaviour. Dobson et al. [24] presented the sorption isotherm measured by Mittlestead and Liu [38], who measured the water sorption vs. activity of Nafion 211 and came up with a model similar to that of Springer et al. [30]. However, they did not account for elevated temperatures (<100 °C) and the phase of water (vapour and liquid).

Fuel cell performance decreases with increasing temperatures beyond 100 °C, which is due to the issue mentioned above of water accumulation and retention at these temperatures. Research into composite Nafion membranes showed the potential of operating Nafion at temperatures up to 120 °C [19]. However, in addition to the polymer modification, the operating conditions, such as the operating pressure, need to be increased with increasing operating temperature to enable the correct relative humidity and, therefore, water sorption [39,40,41]. For the first time, this work attempts to assess the performance of a PEFC equipped with a Nafion 211 membrane under elevated temperatures (80 °C, 100 °C, and 120 °C) and various relative humidity (40–100%). Membrane performance was assessed in a single PEFC cell. The experimental results were used to extend the model of Nafion 211 to adapt to elevated temperatures and enable the investigation of optimum operating pressures and the relative humidity at these operating temperatures.

## 2. Experimental Method and Model Formulation

### 2.1. Materials and Experimentation

The membrane electrode assemblies (MEAs) were prepared as follows: A 5 cm^2^ gas diffusion electrode (GDE) (Johnson Matthey, Swindon, UK GDE with 0.2 mg cm^−2^ Pt loading) was used for the anode and the cathode. The anode GDE, the cathode GDE, and the membrane (Nafion 211, Chemours, Willmington, DE, USA) were assembled by hot pressing at 130 °C for 4 min, with 2 of those minutes being under compression weight of 0.125 tonnes.

The MEA was then assembled into a single cell and tested using a fuel cell test station (850e, Scribner Associates, Southern Pines, NC, USA). Hydrogen and air with controlled temperature and relative humidity were applied to the anode and cathode sides. It is important to note that the relative humidity conditions were kept equal for both electrodes during the test. Electrochemical measurements, including polarisation curve and electrochemical impedance spectroscopy (EIS), were performed on the cell at varied operating conditions, as detailed in Table 1. EIS was conducted at 0.6 V by applying an AC signal with a frequency sweep from 0.1 Hz to 10 kHz and an amplitude of 10% of the DC voltage. The EU harmonisation protocol for cell break-in was conducted before performing the measurements [42].

### 2.2. Modelling

COMSOL Multiphysics 5.6 was used to model a 1D representation of a fuel cell; the focus of this model is to understand the interplay between the membrane water management and its performance at high temperatures, measured by the total performance of the fuel cell. The modelling approach adopted in this study is explained in the flow diagram shown in Figure 1. Experimental data for the lumped high-frequency resistance (HFR) was used to calculate the membrane water content (Lambda (λ)). The model is run with maximum activity, the calculated Lambda is compared to the experimental Lambda, and once an acceptable error is achieved (<5%), the model IV curve data is compared to the experimental IV curve results.

The model was made one-dimensional to isolate planner dimensional effects, see Figure 2. The main three assumptions used for the model are the following: (i) the membrane electrolyte is impermeable to gases, (ii) the membrane is homogeneous in the in-plane domain, and (iii) the gases are ideal. The model accounts for the transport of chemical species in the porous media (hydrogen and water vapour on the anode side; oxygen, water vapour and nitrogen on the cathode side), charges (ions and electrons), heat, liquid water in the porous media (represented by water saturation) and dissolved water in the membrane phase (represented by Lambda). Below are the conservation equations used to model the transport of each of the quantities mentioned above.

#### 2.2.1. Model Formulation

##### Transport of Chemical Species

The following equations describe the general governing physics of the transport of a concentrated species, including convective, diffusive, and reactive sources.
(1)$$\nabla \cdot N_{j}=S_{j}$$

**N***_j_* and S*_j_* are the mass flux and source terms of species *j* (i.e., O_2_, H_2_O, or N_2_), respectively. **N***_j_* is obtained using the Maxwell–Stefan equation.
(2)$$N_{j}=-\rho \omega_{j}\sum_{k}D_{jk}^{eff}\frac{M}{M_{k}}\;\left(\nabla \omega_{k}+\omega_{k}\frac{\nabla M}{M}\right)$$
(3)$$\rho =\frac{pM}{RT}$$

The effective diffusivity of the gases into the gas diffusion layers is given as follows [43,44]:(4)$$D_{jk}^{eff}=0.008e^{4.81\varepsilon }D_{jk}$$
where *ε* is the porosity and the *D_jk_* is the diffusivity. On the other hand, the effective diffusivities within the catalyst layers were estimated using the well-known Bruggeman approximation as follows:(5)$$D_{jk}^{eff}=\varepsilon_{cl}^{1.5}D_{jk}$$

The diffusivities were corrected for pressure and temperature using the following equation [45]:(6)$$D_{ij}=D_{ij}\left(T_{0},p_{0}\right)\frac{p}{p_{0}}{\left(\frac{T}{T_{0}}\right)}^{1.5}$$

The reference pressure $p_{0}$ is 1 atm, and the values for the reference temperatures $T_{0}$ were taken from Berning et al. [45]. The source term in Equation (1) is the consumption/production rate of oxygen, hydrogen, and water vapour, as follows:(7)$$S_{O_{2}}=-\frac{IM_{O_{2}}}{4F}$$
(8)$$S_{H_{2}}=-\frac{IM_{H_{2}}}{2F}$$
(9)$$S_{H_{2}O}=I\left(\frac{n_{d}}{F}+\frac{M_{H_{2}O}}{2F}\right)$$
where I is the volumetric current density (A/m^3^), $M_{O_{2}}$, $M_{H_{2}}$, and $M_{H_{2}O}$ are the molecular weight of oxygen, hydrogen, and water, respectively, F is Faraday’s constant and $n_{d}$ is the net-drag coefficient that is calculated as follows:(10)$$n_{d}=\frac{2.5\lambda }{22}$$
where λ is water content.

##### Transport of Electronic and Ionic Charge

The following equations govern the transport of electrons and protons in the electrically and ionically conductive components:(11)$$\nabla \left(-\sigma_{s}^{eff}\nabla \phi_{s}\right)=\nabla .i$$
where the solid phase potential is $\phi_{s}$ and the membrane phase potential is $\phi_{m}$ and i is the current density (A/m^2^). $\sigma_{s}^{eff}$ and $\sigma_{m}^{eff}$ are the effective electrical conductivity of the solid phase and effective membrane conductivity, respectively. The gas diffusion layer effective electrical conductivity, $\sigma_{GDL}^{eff}$. The electrical conductivity of the catalyst layer is calculated as follows:(12)$$\sigma_{cat}^{eff}=\sigma_{GDL}^{eff}{\left[\left(1-\varepsilon_{cat}\right)\left(1-\nu_{naf}\right)\right]}^{1.5}$$
where $\varepsilon_{cat}$ is the porosity of the catalyst layer and $\nu_{naf}$ is the volume fraction of the ionomer phase in the catalyst layer. The “water transport in membrane phase” subsection defines the effective membrane conductivity. The activation overpotential η for the anode ($\eta_{a}$) and the cathode ($\eta_{c}$) electrodes is obtained as follows:(13)$$\eta_{a}=\Phi_{s}-\Phi_{m}$$
(14)$$\eta_{c}=\Phi_{s}-\Phi_{m}-E$$
where E is the equilibrium potential and is given by the following:(15)E=E0+RT2FlnPH2.PO212
where $E_{0}$ is the reference equilibrium potential at ambient temperature (1.23 V), $P_{H_{2}}$ and $P_{O_{2}}$ are the partial pressures of hydrogen and oxygen, respectively. The above activation overpotential is required to calculate the anodic ($I_{a}$) and cathodic ($I_{c}$) volumetric current densities through the following Butler–Volmer equations:(16)$$I_{a}=I_{o,a}\frac{C_{H_{2}}}{C_{H_{2}}^{ref}H_{H_{2}}}\left(e^{\frac{-\left(1-\alpha_{a}\right)F\eta_{a}}{RT}}-e^{\frac{\alpha_{a}F\eta_{a}}{RT}}\right)\;$$
(17)$$I_{c}=A_{Pt}^{eff}i_{o,c}\left(1-s\right)\frac{C_{O_{2}}}{C_{O_{2}}^{ref}H_{O_{2}}}\left(e^{\frac{-\alpha_{c}F\eta_{c}}{RT}}\right)\;$$
where $I_{o,a}$ is the anodic volumetric current density (A/m^3^), $i_{o,c}$ is the cathodic exchange current density, $C_{i}$ is the concentration of the species i ($H_{2}$ or $O_{2}$), and $C_{i}^{ref}$ is the reference concentration for the species i ($H_{2}$ or $O_{2}$), and $H_{i}$ is Henry’s law coefficient for the species i ($H_{2}$ or $O_{2}$). R is the universal gas constant, F is Faraday’s constant, and s is the saturation. Note that the water vapour saturation is accounted for in the Butler–Volmer equation of the cathode electrode (i.e., Equation (18)). The effective area of platinum available throughout the cathode catalyst layer, $A_{Pt}^{eff}$, is given by the following:(18)$$A_{Pt}^{eff}=\frac{A_{pt}*m_{Pt}}{L_{cat}}$$
where $A_{pt}$ is the specific area of platinum, $m_{Pt}$ is the mass of platinum and $L_{cat}$ is the thickness of the catalyst layer.

##### Transport of Heat

The following equation [45] governs the heat transfer in the various components of the model:(19)$$\nabla .\left(k_{s}\nabla T\right)+S_{e}=0$$
where k is the thermal conductivity and $S_{e}$ is the heat (or energy) source term. The source terms for each component are given in Table 2.

##### Water Transport in the Membrane Phase

The dissolved water in the membrane phase (in the membrane and the membrane phase in the catalyst layers) is represented by water content λ, and the following conservation equation governs its physics:(20)$$\nabla \cdot \left(-D_{w}\frac{\rho_{mem}}{EW}\nabla \lambda \right)=\left\{\begin{array}{c} \nabla \left(\frac{n_{d}\sigma_{mem}^{Eff}}{F}\nabla \Phi_{m}\right)\;\;\;\;\;\;in\;CLs\\ \nabla \left(\frac{n_{d}\sigma_{mem}}{F}\nabla \Phi_{m}\right)\;in\;membrane\;electrolyte \end{array}\right.$$
where $D_{w}$ is the water diffusion coefficient, $\rho_{mem}$ is the membrane density, and EW is the equivalent weight of the membrane. The membrane ionic conductivity for 212 Nafion membranes is given as follows [46]:(21)$$\sigma_{mem}=\left(2.0634+1.052\lambda +0.010125\lambda^{2}\right)e^{\left[751.4\left(\frac{1}{303}-\frac{1}{T}\right)\right]}$$
where λ is water content, the conductivity of the membrane phase in the catalyst layers are corrected for as follows:(22)$$\sigma_{mem}^{eff}=\sigma_{mem}{\left(\left(1-\varepsilon_{cat}\right)\nu_{naf}\right)}^{1.5}$$

Boundary condition values of water content at the interfaces between the gas diffusion layers and the catalyst layers are required to solve Equation (20); these boundary conditions water contents ($\lambda_{BC}$) are given using the following equation [46] where the temperature is given in Celsius:(23)$$\lambda_{BC}=\left(1+0.2352a^{2}\right).\left(\frac{T-30}{30}\right).\left(14.22a^{3}-18.92a^{2}+13.41a\right)$$
where a is water activity and is given by the following:(24)$$a=\frac{p_{w}}{p_{s}}$$
where $P_{w}$ is the partial pressure of water vapour and $p_{s}$ is saturation pressure of water vapour and obtained (in atm) as follows:(25)$$\begin{array}{ll} \log_{10}(p_{s})= & -2.1794+0.02953\left(T-273.15\right)-9.1837\times 10^{-5}{\left(T-273.15\right)}^{2}\\ & +1.4454\times 10^{-7}{\left(T-273.15\right)}^{3} \end{array}$$

##### Water Vapour Saturation

The transport of liquid water in the porous media (i.e., the gas diffusion layers and the catalyst layers) is accounted for through the following conservation equation of water vapour saturation, s [47,48]:(26)$$\nabla \cdot \left(\frac{\rho_{w}Ks^{3}}{\mu_{w}}\frac{dp_{c}}{ds}\nabla s\right)=S_{s}$$
where $\rho_{w}$ and $\mu_{w}$ are, respectively, the density and dynamic viscosity of liquid water, K is the absolute permeability, and $p_{c}$ is the capillary pressure, which is given by the following:(27)$$p_{c}=\frac{\sigma_{w}\cos\left(\theta \right)}{\sqrt{\frac{K}{\varepsilon }}}1.417s-2.12s^{2}+1.263s^{3}$$
where $\sigma_{w}$ is the surface tension of liquid water, and θ is the contact angle. $S_{s}$ is the saturation source term and is obtained by the following:(28)$$S_{sat}=\left\{\begin{array}{c} \gamma M_{H_{2}O}\left(1-s\right)\frac{\left(P_{w}-P_{s}\right)}{RT}\;if\;P_{w}>P_{s}\\ \gamma M_{H_{2}O}s\frac{\left(P_{w}-P_{s}\right)}{RT}\;if\;P_{w}\leq P_{s} \end{array}\right.\;$$
where γ is the condensation rate constant, note that the conservation equation of saturation was only applied to the cathodic gas diffusion layer and the catalyst layer (where water is produced) and was not applied to the anode side as water condensation is unlikely.

### 2.3. Boundary Conditions and Numerical Procedure

Dirichlet boundary conditions (i.e., specific values) were prescribed for the chemical species, the temperature, the saturation, and solid-phase potential at the edges of the gas diffusion layers. Likewise, Dirichlet boundary conditions were set for the membrane phase potential and water content at the interfaces between the gas diffusion and catalyst layers. Table 3 lists the boundary conditions used for the model.

The governing equations of the model were solved using COMSOL Multiphysics^®^ 5.6 solver. The computational domain was discretised and refined (especially at interfaces and boundaries) until a mesh-independent solution was obtained; the number of elements that achieved this was 2615. Table 4 shows the parameters used in the model.

## 3. Results

Figure 3 shows the experimental polarisation and EIS results for the MEA testing at different operating temperatures and reactants’ relative humidity. The trend for the data at 80 °C (Figure 3a,b) shows a decrease in performance as the relative humidity increases; this is due to the accumulation of water at high current density, which causes an increase in the mass transport polarisation. The EIS taken at 0.6 V shows that the ohmic resistance is constant for all relative humidity conditions. Towards the low-frequency end of the impedance data, there is a clear shift in the total cell resistance as the mass transport effects become onset. The relationship is inverse due to the oxygen concentration decreasing with increasing relative humidity and the minor onset of flooding behaviour.

The results at 100 °C are shown in Figure 3c,d. Generally, the cell shows an increase in the maximum performance achieved: the average performance for 80 °C, 0.6 V was 1200 mA/cm^2^ and the performance at 100 °C, 0.6 V was 1400 mA/cm^2^. The trend here is reversed compared to that seen at 80 °C, with the highest performance is achieved at high relative humidity between 60–100%, while at 40% relative humidity, the cell shows signs of dehydration with a significant increase in the cell resistance compared to the 100% relative humidity condition. The EIS results (Figure 3d) confirm the increase in the ohmic resistance of the cell with reducing relative humidity. Moreover, it is important to note at the top left of the polarisation curve the reduction in OCV for the cell compared to that achieved at 80 °C.

The results at 120 °C show a similar performance trend to that seen at 100 °C but with a reduced total cell performance. The EIS results in Figure 3f demonstrate the further increase in ohmic resistance compared to the other temperatures due to dehydration. Again, there is a further reduction in OCV than at other temperatures, which can be related to reducing the Nernst potential and increasing hydrogen crossover with increasing temperatures.

Furthermore, the EIS results shown in Figure 3 were used to obtain a relationship for the membrane resistance at the different operating conditions [52,53,54,55]. The typical equivalent circuit method [56] was not used as it fails to account for the complex resistances in the catalyst layer, contact resistances between fuel cell components, and separate ionic and electronic resistances. Work by Pickup’s group [52,53,54] showed that using the spectra’s real intercept at low and high-frequency, it is possible to deconvolute the electronic and ionic resistances within the cell. The ionic resistance of the cell comprises the membrane and the ionomer in the catalyst layer. Therefore, their method was employed to generate the ionic resistance values in Table 5 using Equation (29). The Lambda values were then calculated using Equation (23), from both the HFR value (uncorrected) and the ionic resistance calculated using Pickup’s method (corrected), and the difference between the calculated Lambda values is displayed in Figure 4.
(29)$$\frac{1}{R_{T}}=\frac{1}{R_{ionic}}+\frac{1}{R_{electronic}}$$

While it is helpful to understand the specific resistances within the ohmic region, these resistances are not accounted for in the model; therefore, using a lumped parameter fits the model’s design. However, knowing the real resistance associated with the water content of the membrane, and not the inflated lumped parameter, allows this work to be valuable at other levels.

Figure 4 shows the difference between corrected and uncorrected Lambda; the graph shows a relative shift down across all the activities and temperatures. The electronic resistance is known to vary with clamping pressure, temperature, and humidity. This method shows that the true Lambda can be described accurately from experimental data, and that the electronic resistance is relatively constant across the range of relative humidity. Moreover, separating the electronic and ionic charges in the lumped parameter model is unnecessary as it is clear that the largest portion of the variance in Lambda comes from the ionic resistance value. It does mean that when using a lumped parameter model, larger activities should be used to account for the electronic resistance such as contact resistances that are unaccounted for otherwise. Therefore, the maximum activity can be used as a clear fitting parameter even though it is inflated. It is possible to back-calculate the true ionic resistance once the inflated activity is considered. The corrected values can be seen to agree with previous work’s reported values for ex-situ measurements [24,54,58]. It is expected that the activity values obtained here would be inflated slightly because of the higher density of water vapour and sensible heat at higher pressures, meaning that there could be more water in the gas phase occupying the same space.

Table 6 shows the relative increase between the corrected and uncorrected Lambda values, which indicate the difference when assessing the maximum activity. The correction gets more significant as temperature increases due to the increase in the cell resistance, meaning that the HFR represents a slightly higher proportion of electronic to ionic resistance as temperature increases.

Figure 5 shows the equation expressed by Dobson et al. [46]. The relationship was defined for activates up to 1, representing 100% relative humidity. However, pressure affects the vapour density, causing it to rise, which would increase Lambda, as this is defined as the moles of water per mole of acid group in the sidechain of the polymer. Hence, the relationship was extended to include the range of measured values.

Figure 6 shows the Lambda values calculated from the experimentally determined high-frequency resistance (HFR) compared to the water content values calculated from the model. The maximum water activity was used as a fitting parameter. The measured water content was taken at 0.6 V, so similarly, the model used 0.6 V. As discussed above, the lumped Lambda shown in this figure is slightly higher than the actual Lambda.

Table 7 shows the fitting parameter (i.e., water activity) values used to generate the corrected water content within the membrane. The higher temperatures show a reduction in the maximum water activity, which is expected as higher temperatures increase the saturation pressure of water vapour.

Figure 7 demonstrates the model’s accuracy in predicting the polarisation curve using the derived membrane properties against the experimental results. The model shows excellent representation of the experimental results in the ohmic region at all temperatures. At 120 °C, the model predicts the slope of the ohmic region very well. Due to the temperature increase, the ohmic resistance dominates the majority of the polarisation curve, while reducing the effect of mass transport.

## 4. Discussion and Model Predictions

This work shows that the empirical relationship derived for Nafion 211 [24] for Lambda vs. water activity is valid up to 120 °C with high relative humidity. The model shows good agreement with the experimental results in the ohmic region of the polarisation curve. The shape of the water distribution shown in Figure 6 changes between 80 °C and 100 °C or 120 °C and is probably due to several factors. The inlet gas stream’s relative humidity and general thermodynamic conditions govern the membrane’s water content stability. As shown in the steam table, the enthalpy of vapourisation decreases with temperature, and the liquid enthalpy increases with temperature, making it much easier to produce water vapour at higher temperatures. Pressure has a different effect that affects the quality of steam; the higher the pressure, the higher the boiling temperature, which changes the ratio of liquid to vapour. The backpressure in the streams was maintained at 3 bar, and the saturation pressure (Ps) is temperature dependent.

Table 8 shows that the oxygen content at 80 °C is approximately 2.5 times greater than the value at 120 °C. Increasing the temperature of a Nafion membrane, which requires high hydration levels, causes a significant issue in terms of cathode oxygen content. The graph also shows that the water content doubles from 80 °C to 100 °C and then doubles again from 100 °C to 120 °C; therefore, this is probably why the high-temperature tests typically have lower performance than the lower temperature tests.

Performance is hampered at high temperatures by the proportion of the gas stream occupied by water; this water takes up space that could be occupied by oxygen and increases the likelihood of flooding in the cell. It also means that the mass flow rate of water required to keep the membrane fully hydrated is much greater than at 80 °C, increasing the likelihood of dehydration. Especially considering temperatures above the boiling point of water, higher pressure testing is required to improve the oxygen content of the cathode gas stream and achieve the desired relative humidity.

The impedance analysis of the 80 °C and 100 °C agrees nicely with the literature [52,53,54,55]. However, the high relative humidity data in Figure 3d,f shows the development of a second semi-circle, indicating that there is a shift in the dominating resistance at 0.6 V, which would indicate that the analysis does not strictly apply or that the mass transport losses have increased to such a point that they are starting to dominate in this voltage region. This is apparent at 120 °C.

The flooding physics used in the model does not represent the observed behaviour [59,60]. The model overestimates the mass transport region of the polarisation curve; therefore, dehydration and greater mass transport resistance physics are a better fit. It is confirmed with the EIS showing the development from a single semi-circle, indicating dominating ohmic resistance, to two semi-circles, indicating a mix of mass transport and ohmic resistance. The intercept between the two semi-circles represents the limit of the ohmic resistance, and the low-frequency intercept now shows the mass transport resistance.

The Lambda is stable at 80 °C. This is not seen at higher temperatures, and there is a more dramatic drop off in Lambda as RH decreases. It shows that lower humidity conditions will affect performance significantly; the key is hydration, but as discussed above, the increase in temperature will also change the oxygen available to react.

The model can predict the fuel cell’s performance at many combinations of different operating conditions; Table 9 shows the current density obtained at a cell voltage of 0.6 V and varied temperature, relative humidity, and pressure. It can be noted that the performance is similar between 80 °C and 100 °C. At 120 °C, the performance is significantly hindered, most likely because of the oxygen content mentioned in the above paragraph. 0.6 V was chosen as the performance parameter as it is closely tied to the membrane performance. The colour shading in the table indicates high performance (green) and low performance (red). The empty spaces in Table 9 at 1 bar pressure represent the fact that it is impossible to achieve those relative humidity conditions at 1 bar pressure as the saturation pressure exceeds the total pressure. Hence, we see a significant improvement between the 2 bar and 3 bar data.

Figure 8 shows the model prediction data in surface plots, blue in the surface plot represents high performance, and yellow represents low performance. Generally, higher pressure and lower temperatures are better for the ohmic region. It appears that higher temperatures generally decrease performance; however, this is only true at the ohmic region; it can be seen from the polarisation curves that the lower temperatures appear more affected by the mass transport limitations than the higher temperatures, although 120 °C was greatly affected by the membrane’s increased resistance and reduced performance.

## 5. Conclusions

This work shows that empirical relationships that relate activity to Lambda (λ) and ionic conductivity for membrane materials can apply to higher temperature scenarios (≥100 °C). The evaluation of the activity and Lambda should be the critical fitting parameters when creating and validating a model, as this is the crucial difference between regular operation and intermediate temperature scenarios. When activity is below 80%, the relationship begins to break down. Measurements of water sorption in the range of interest (80–120 °C) are needed; however, creating an experiment to do this could prove challenging, especially as tight temperature and pressure control are needed to achieve accurate measurements.

The key determinant in the ohmic region of performance is membrane hydration, and it can be seen that as this decreases, the performance too will decrease. However, it is not all of the resistance, and when the relative humidity and temperature are high, other resistances can dominate. This means that a careful impedance analysis can be used to obtain the corrected values; this is especially important to agree with sorption models as lumped parameters will give inflated Lambda values.

The model has shown that a general pressure increase improves performance when considering the amount of available oxygen. The highest performance achieved experimentally was at 100 °C because the increased temperature significantly improves mass transport, especially in eliminating the flooding phenomena. There is an advantage to increasing the pressure of fuel cells that has yet to be explored; the limit of 80 °C could be easily breached if more work is performed to understand high-pressure scenarios. This work shows that there is merit in the high temperature operation of fuel cells made from Nafion. The following are some key parameters that need to be balanced correctly: water content (Lambda), pressure, and oxygen concentration. Simply turning up the temperature shows no obvious benefit; however, this work shows that there could be value in increasing the operating pressure beyond 3 bar, as the trend shows that greater pressure leads to easier water balancing without sacrificing available oxygen.

This model could be improved by creating a method to measure the empirical water sorption relationship of the membrane at temperatures between 90 °C and 120 °C and pressures from atmospheric to 3 bar. The materials in fuel cells are constantly improving, and fundamental work holds modelling back from being implemented more effectively in design.

## Figures and Tables

**Figure 1 membranes-12-00430-f001:**
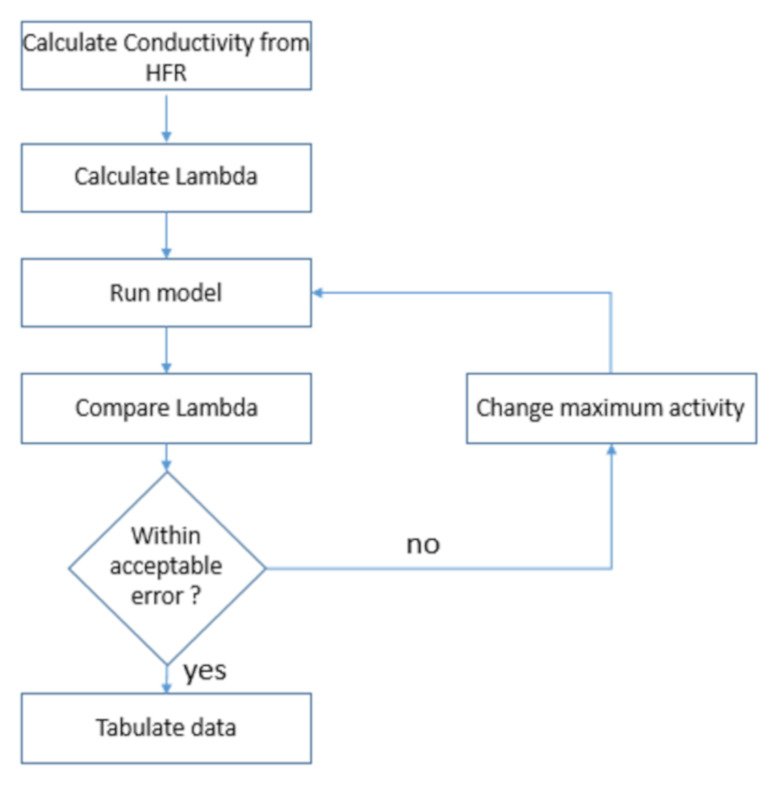
Model development methodology flow chart.

**Figure 2 membranes-12-00430-f002:**
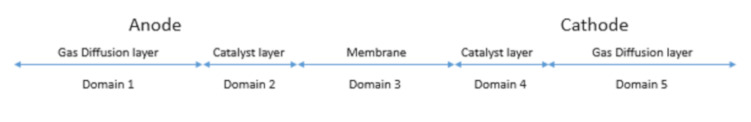
The geometry of the one-dimensional PEFC model.

**Figure 3 membranes-12-00430-f003:**
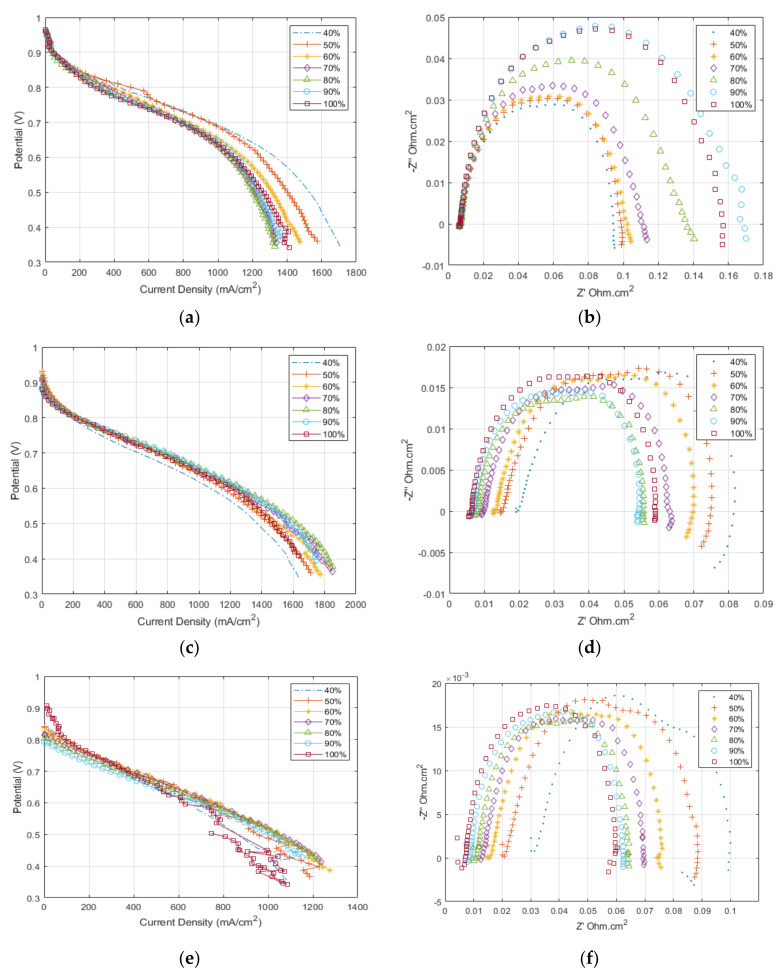
Polarisation curves at different temperatures (**a**) 80 °C, (**c**) 100 °C, and (**e**) 120 °C and Impedance spectroscopy at 0.5V over different temperatures (**b**) 80 °C, (**d**) 100 °C, and (**f**) 120 °C, data is separated by relative humidity.

**Figure 4 membranes-12-00430-f004:**
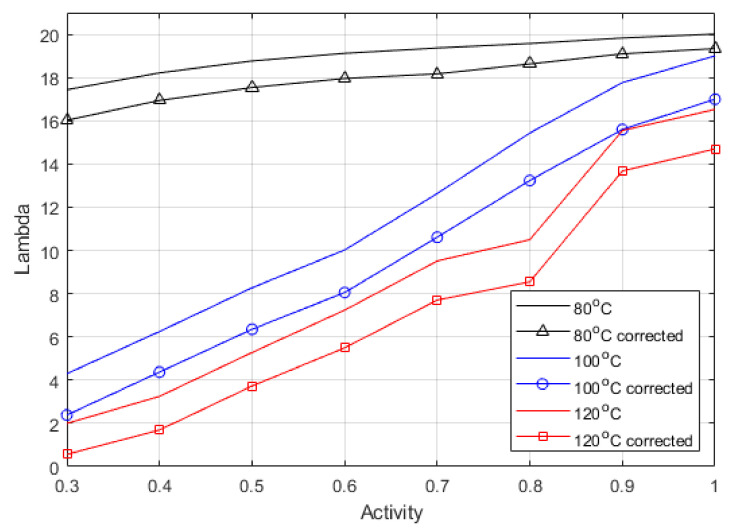
Corrected vs. uncorrected Lambda values from deconvolution impedance analysis.

**Figure 5 membranes-12-00430-f005:**
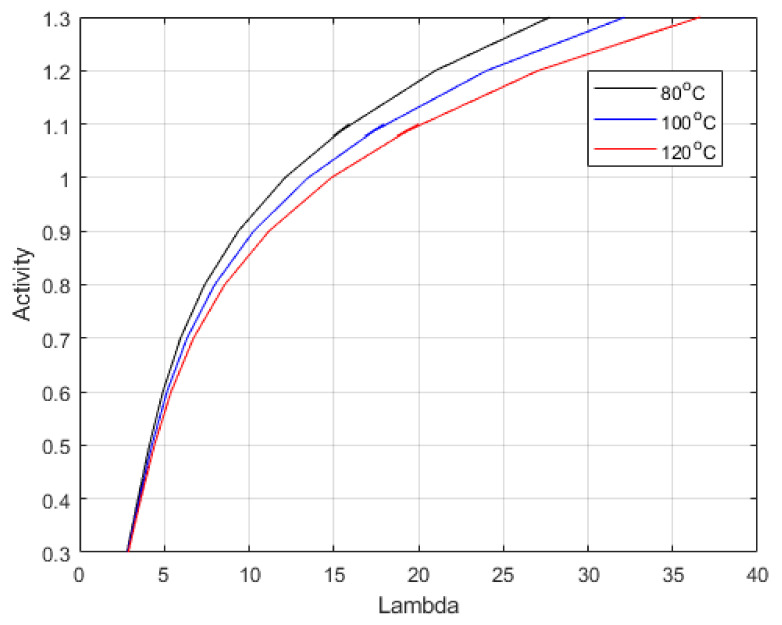
Predicted Lambda vs. water activity (a) from Equation (23) at varied temperatures.

**Figure 6 membranes-12-00430-f006:**
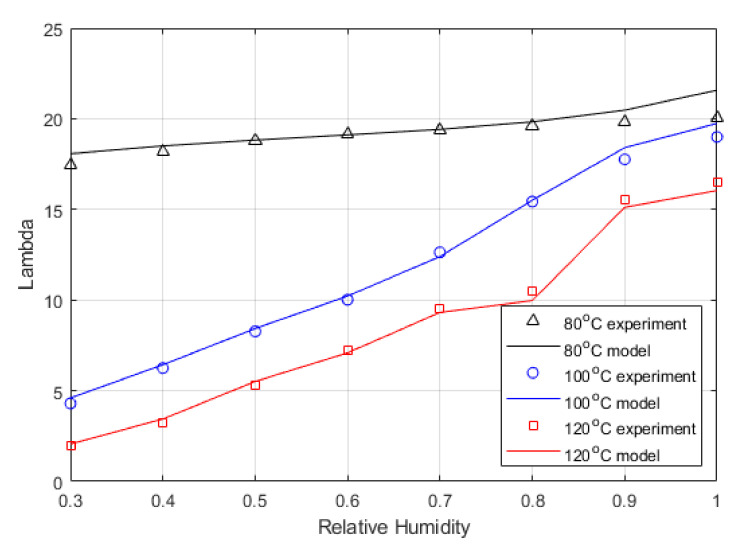
Measured (from HFR impedance spectroscopy) and predicted water content from the developed model versus relative humidity.

**Figure 7 membranes-12-00430-f007:**
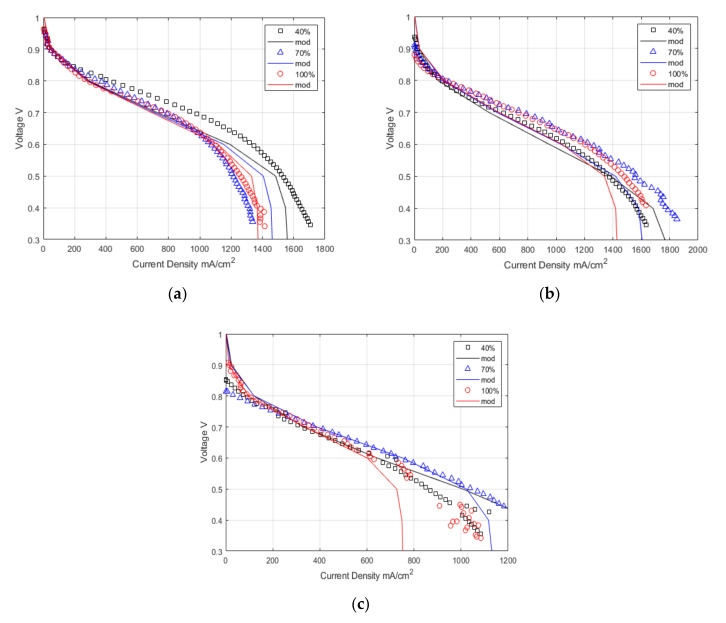
Experimental and modelling data for polarisation curves at (**a**) 80 °C, (**b**) 100 °C, (**c**) 120 °C, and 40%, 70%, and 100% Relative humidity.

**Figure 8 membranes-12-00430-f008:**
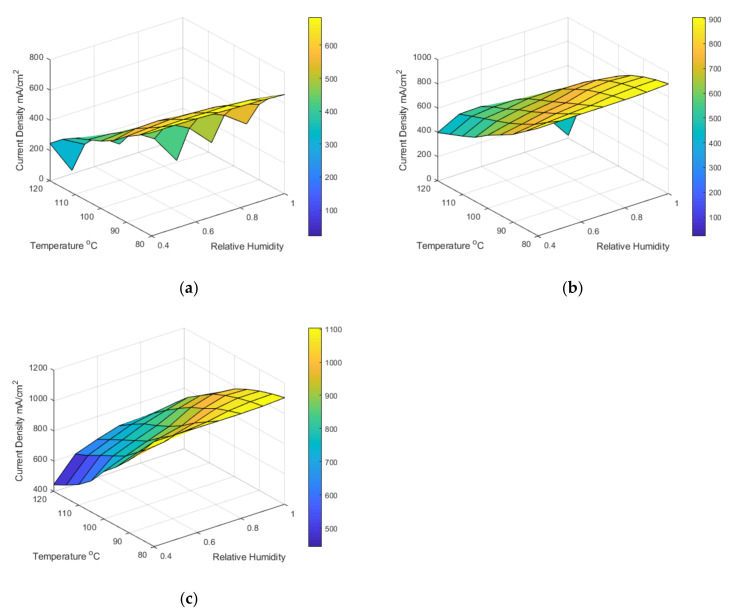
Surface plots of model current density prediction data at 0.6 V across 40–100% RH, 80–120 °C, and (**a**) 1 bar, (**b**) 2 bar, and (**c**) 3 bar.

**Table 1 membranes-12-00430-t001:** MEA test conditions in the PEMFC single cell.

Parameters	Anode	Cathode
Fuel	Hydrogen	Air
Temperature (°C)	80	80
Flow rate (mL min^−1^)	150	500
Relative humidity (%)	40/50/60/70/80/90/100	40/50/60/70/80/90/100
Back pressure (bar)	2	2

**Table 2 membranes-12-00430-t002:** Heat transfer source terms.

Component	Source Term
Gas diffusion layers	$\sigma_{GDL}\Phi_{s}^{2}$
Catalyst layers	$-\eta_{c/a}I_{c/a}+\sigma_{CL}^{Eff}\Phi_{s}^{2}+\sigma_{mem}^{Eff}\Phi_{m}^{2}$
Membrane	$\sigma_{mem}\Phi_{m}^{2}$

**Table 3 membranes-12-00430-t003:** Dirichlet boundary conditions used for the model. Note that boundary conditions water contents are calculated using Equation (22).

Variable	Value
Mass fraction of oxygen	0.15
Mass fraction of water vapour at the cathode	0.06
Mass fraction of hydrogen	0.12
Voltage of cathodic terminal	0.3–1 V
Voltage of anodic terminal	0 V
Saturation	0.1
Temperature	80, 100, 120 °C

**Table 4 membranes-12-00430-t004:** Modeling parameters used for the model [45,49,50,51].

Symbol	Value	Units	Description
*A_Pt_*	40	m^2^ g^−1^	Catalyst specific area
*A_s_*	1000	m^−1^	Saturation specific area
*α_c_*	0.8		Cathode charge transfer coefficient
*α_a_*	0.5		Anode charge transfer coefficient
*C* _*H*2_ * ^Ref^ *	5.64 × 10^−5^	mol cm^−3^	Reference concentration for Hydrogen
*C* _*O*2_ * ^Ref^ *	0.85111	mol m^−3^	Reference concentration for Oxygen
*D* _*O*2,*naf*_	8.45 × 10^−10^	m^2^ s^−1^	Diffusivity of Oxygen in Nafion
*EW*	1000	g mol^−1^	Equivalent weight of membrane
*F*	96,485	C mol^−1^	Faraday’s constant
*H* _*H*2_	4.56 × 10^3^	Pa m^3^ mol^−1^	Henry’s constant for H_2_
*H* _*O*2_	0.3125	atm m^3^ mol^−1^	Henry’s constant for O_2_
*I_o,a_*	1.4 × 10^−5^	A cm^−3^	Reference volumetric exchange current density anode
*I_o,c_*	0.5	A cm^−2^	Reference exchange current density cathode
*L_cat_*	15	μm	Catalyst layer thickness
*L_GDL_*	265	μm	Gas diffusion layer thickness
*L_mem_*	25	μm	Membrane thickness
*M* _*H*2_	2	g mol^−1^	Molecular weight hydrogen
*M* _*H*2*O*_	18	g mol^−1^	Molecular weight water
*M* _*N*2_	28	g mol^−1^	Molecular weight nitrogen
*M* _*O*2_	32	g mol^−1^	Molecular weight oxygen
*m_Pt_*	0.4	mg cm^−2^	Platinum loading
*Net drag*	1		Electro-osmotic drag Coefficient
*P*	3	Bar	Pressure
*ε_cat_*	0.48		The porosity of the catalyst layer
ε_GDL_	0.45		The porosity of the gas diffusion layer
*ν_sol_*	0.48		The volume fraction of the solid phase
*R*	8.314	J mol^−1^ K^−1^	Universal ideal gas constant
*ρ_mem_*	2000	kg m^−3^	Membrane Density
*σ_GDL_*	100	S m^−1^	Gas diffusion layer effective electrical conductivity
*ν_naf_*	0.25		Electrolyte volume fraction
*ρ_w_*	1000	kg m^−3^	Density of water
*μ_w_*	4.05 × 10^−4^	Pa s	Viscosity of water
*K*	1 × 10^−13^	m^2^	Absolute permeability
*σ_w_*	0.0644	N m^−1^	Surface tension
*θ*	105	degree	Contact angle

**Table 5 membranes-12-00430-t005:** HFR intercept (Ω.cm^2^) compared to the calculated ionic resistance from Pickup’s group’s work [52,53,54,55,57].

Activity\Temp	80			100			120		
	HFR	LFR	Ionic	HFR	LFR	Ionic	HFR	LFR	Ionic
0.3	0.0077	0.0958	0.0086	0.02750	0.0879	0.0440	0.0451	0.1213	0.0887
0.4	0.0075	0.0946	0.0081	0.01970	0.0814	0.0271	0.0313	0.0994	0.0504
0.5	0.0072	0.0988	0.0078	0.01510	0.0747	0.0194	0.0207	0.0885	0.0280
0.6	0.0071	0.1021	0.0076	0.01250	0.0697	0.0155	0.0155	0.0750	0.0200
0.7	0.0070	0.1117	0.0075	0.00987	0.0635	0.0118	0.0119	0.0695	0.0146
0.8	0.0069	0.1365	0.0073	0.00799	0.0555	0.0094	0.0108	0.0643	0.0132
0.9	0.0068	0.1667	0.0071	0.00686	0.0542	0.0079	0.0072	0.0625	0.0082
1	0.0067	0.1569	0.0070	0.00637	0.0590	0.0072	0.0067	0.0597	0.0076

**Table 6 membranes-12-00430-t006:** Increase in Lambda between corrected and uncorrected values.

Activity\Temperature (°C)	80	100	120
0.3	1.087	1.801	3.537
0.4	1.075	1.429	1.917
0.5	1.070	1.302	1.416
0.6	1.064	1.244	1.317
0.7	1.065	1.190	1.234
0.8	1.050	1.165	1.228
0.9	1.038	1.138	1.136
1	1.034	1.118	1.124
**Average**	1.061	1.193	1.243

**Table 7 membranes-12-00430-t007:** The fitted water activity used in the model for each relative humidity and temperature.

Relative Humidity (%)	80 °C	100 °C	120 °C
**40**	1.2	0.92	0.4
**50**	1.2	0.97	0.8
**60**	1.2	1.02	0.9
**70**	1.2	1.07	1
**80**	1.2	1.15	1.07
**90**	1.2	1.2	1.17
**100**	1.2	1.2	1.17

**Table 8 membranes-12-00430-t008:** Cathode oxygen content vs. temperature at 100% relative humidity.

T (°C)	P (BAR)	PSAT (BAR)	MOL% H_2_O	MOL% AIR	MOL% O_2_
**80**	3	0.4741	0.1580333	0.841967	0.176813
**100**	3	1.0142	0.3380667	0.661933	0.139006
**120**	3	1.9867	0.6622333	0.337767	0.070931

**Table 9 membranes-12-00430-t009:** Prediction of current density at 0.6 V across 80–120 °C and 1–3 bar, highest performance in green and lowest performance in red.

**1 Bar**	**Relative Humidity (%)**						
**Temp**	**40**	**50**	**60**	**70**	**80**	**90**	**100**
80	685.09	684.97	681.64	676.23	669.96	664.3	655.08
85	621.86	629.1	631.21	629.04	629.97	619.94	596.52
90	566.61	576.91	587.01	582.21	574.27	548.58	498.74
95	493.93	527.04	531.43	516.24	490.46	427.75	322.89
100	444.94	457.05	453.18	420.66	343.06	200.73	
105	408.49	400.74	364.52	273.97	83.548		
110	373.05	337.49	239.87				
115	319.62	239.68					
120	246.79	20.678					
**2 bar**	**Relative Humidity (%)**						
**Temp**	**40**	**50**	**60**	**70**	**80**	**90**	**100**
80	890.38	892.43	892.58	892.13	892.71	896.44	906.73
85	808.85	823.93	835.95	846.15	867.28	878.74	882.85
90	725.82	750.71	782.39	800.42	837.18	853.03	850.15
95	664.39	695.41	731.42	754.08	801.9	817.4	805.25
100	604.44	645.36	682.46	706.4	749.71	768.19	743.05
105	530.43	606.66	641.71	662.57	695.66	693.59	650.45
110	482.65	569.24	589.24	617.23	619.51	598.09	524.8
115	440.48	531.78	549.92	551.3	520.49	455.1	332.87
120	397.65	492.16	496.42	457.75	372.38	233.54	22.988
**3 bar**	**Relative Humidity (%)**						
**Temp**	**40**	**50**	**60**	**70**	**80**	**90**	**100**
80	1094.9	1095.2	1093.8	1092.4	1092.3	1095.5	1104.1
85	981.72	996.98	1025.3	1038.5	1065	1078.8	1083.1
90	879.83	922.59	948.25	985.87	1034.7	1056.8	1056.3
95	790.49	841.61	888.99	922.01	988.58	1028.4	1022.2
100	713.93	769.55	821.98	871.82	950.97	991.88	978.71
105	606.23	724.52	779.65	831.45	897.73	936.11	914.85
110	536.83	682.47	737.27	796.97	839.44	877.97	844.75
115	487.52	636.02	694.39	748	773.91	796.64	748.04
120	443.53	599.46	650.09	692.55	697.76	695.62	624.37

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
