# Peer review of "Intermediate Temperature PEFC’s with Nafion^®^ 211 Membrane Electrolytes: An Experimental and Numerical Study"

_membranes, 2022, doi:10.3390/membranes12040430_

Round 1
Reviewer 1 Report
In this manuscript, the performance of Nafion 211 at elevated temperatures up to 120 ̊oC was analyzed by using an experimentally validated model, so as to investigate complex systems to gain further information that is difficult to get experimentally. The highest performance reported is at 100°C, 100% relative humidity, and at a higher pressure of 3 bar.
I consider the content of this manuscript will definitely meet the reading interests of the readers of the Membranes journal. However, the manuscript still needs certain improvement, and it could be a comprehensive work after revision.
Therefore, I suggest giving a minor revision and the authors need to clarify some issues or supply some more data to enrich the content. In particular, the authors should pay special attention to the grammar errors. I will point out some of them, but not all of them.
- Abstract and Introduction
- Pay attention to the grammar issues, and I suggest double-checking the whole main text. I just point out some examples here:
Line 15, ‘The model showed that the best membrane performance comes from a low temperature of operation of 100°C but much better performance with higher pressure 3 bar.’
Line 95, ‘while Nafion 117 is is 183 μm thick and made by extrusion process.’
Line 293, ‘Moreover, it is important to note the reduction in the open-circuit voltage (OCV) for the cell compared to that achieved at 80 ̊C.’ and so on.
- For the Keywords, it is still missing and has to be supplied.
- Line 34, ‘The main role of the membrane is to separate electronic and ionic charges and provide an impermeable barrier to the gases of the anode and cathode. PFSA membranes require high hydration to conduct protons’
It is better to also introduce the chemical structure of Nafion in combination with the properties/functions. Although Line 58 refers to the backbone and functional groups, the hydrophobic/hydrophilic properties and how the ion-conducting channels are not mentioned at all.
For example, Nafion/PFSA is composed of the hydrophobic PTFE backbone and the hydrophilic sulfonic acid group. The PTFE backbones provide the good chemical and mechanical stability, while the phase separations between hydrophobic and hydrophilic domains when the membrane is hydrated provide the ion transportation channels [Solid State Ionics 319 (2018): 110-116; RSC advances 7.50 (2017): 31164-31172]. That is why dehydration reduces the conductivity of PFSA membranes so significantly. The membrane should have a high species selectivity, which enables protons to pass, but prevent the crossover electro-active species (as an effective barrier layer) [Electrochimica Acta 378 (2021): 138133].
- Line 63, ‘because relaxation increases porosity and decreases the stability of the membrane.’
PFSA membranes are not porous membranes, so where does the porosity come from in PFSA membranes? This issue should be further clarified.
- Experimental method and model formulation
Line 117, ‘The anode GDE, the cathode GDE, and the membrane (Nafion 211) were assembled by hot pressing at 130 °C, for 4 min’
Whether Nafion membrane has undergone pretreatment, such as hydrogen peroxide solution and dilute sulfuric acid solution [Solid State Ionics 319 (2018): 110-116]. If yes, there should be a related and detailed description.
- Results and discussion
- Line 307, ‘ Using the ionic resistance of the cell, the actual water content of the membrane can be determined.’
What is the meaning of ‘the ionic resistance of the cell’? Normally, only the PFSA membrane is ion-conducting, and none of the other components in the fuel cell should be ion-conducting.
- The caption of Table 6 should be ‘Table 6: Increase of lambda between corrected and uncorrected values’. Why use all capital letters suddenly here? It should be avoided. The same applies to Table 8.
- Discussion
Line 373, ‘This work shows that the empirical relationship derived for Nafion 211 for water content vs. water activity is valid up to 120°C with high relative humidity.’
What is the probable reason that after 120 oC the relationship is no longer valid? Is it due to the degradation of membranes at high temperatures (thermal stability issues)? There should be some more details.
Author Response
I would like to thank the reviewer for their comments, I have addressed the following comments in order; my text is red to be clear.
In this manuscript, the performance of Nafion 211 at elevated temperatures up to 120 ̊C was analyzed by using an experimentally validated model, so as to investigate complex systems to gain further information that is difficult to get experimentally. The highest performance reported is at 100°C, 100% relative humidity, and at a higher pressure of 3 bar.
I consider the content of this manuscript will definitely meet the reading interests of the readers of the Membranes journal. However, the manuscript still needs certain improvement, and it could be a comprehensive work after revision.
Therefore, I suggest giving a minor revision and the authors need to clarify some issues or supply some more data to enrich the content. In particular, the authors should pay special attention to the grammar errors. I will point out some of them, but not all of them.
- Abstract and Introduction
- Pay attention to the grammar issues, and I suggest double-checking the whole main text. I just point out some examples here:
The grammar of the whole paper has been addressed, thank you for picking these examples out but you were right there were many others that I had missed.
Line 15, ‘The model showed that the best membrane performance comes from a low temperature of operation of 100°C but much better performance with higher pressure 3 bar.’
Line 95, ‘while Nafion 117 is is 183 μm thick and made by extrusion process.’
Line 293, ‘Moreover, it is important to note the reduction in the open-circuit voltage (OCV) for the cell compared to that achieved at 80 ̊C.’ and so on.
These were corrected as advised.
- For the Keywords, it is still missing and has to be supplied.
Keywords have been added as advised. Keywords includ:
Polymer electrolyte fuel cells, proton exchange membrane, fuel cell modelling, Nafion, intermediate temperature
- Line 34, ‘The main role of the membrane is to separate electronic and ionic charges and provide an impermeable barrier to the gases of the anode and cathode. PFSA membranes require high hydration to conduct protons’
It is better to also introduce the chemical structure of Nafion in combination with the properties/functions. Although Line 58 refers to the backbone and functional groups, the hydrophobic/hydrophilic properties and how the ion-conducting channels are not mentioned at all.
For example, Nafion/PFSA is composed of the hydrophobic PTFE backbone and the hydrophilic sulfonic acid group. The PTFE backbones provide the good chemical and mechanical stability, while the phase separations between hydrophobic and hydrophilic domains when the membrane is hydrated provide the ion transportation channels [Solid State Ionics 319 (2018): 110-116; RSC advances 7.50 (2017): 31164-31172]. That is why dehydration reduces the conductivity of PFSA membranes so significantly. The membrane should have a high species selectivity, which enables protons to pass, but prevent the crossover electro-active species (as an effective barrier layer) [Electrochimica Acta 378 (2021): 138133].
Thank you for this comment, I have included information about the basic structure, it was extensively modelled by mologin D et al, they showed the structure is consistent with the pore and chamber hypothesis.(see lines 43-48) This also confirms the ‘porosity’ issue below..
- Line 63, ‘because relaxation increases porosity and decreases the stability of the membrane.’
PFSA membranes are not porous membranes, so where does the porosity come from in PFSA membranes? This issue should be further clarified.
PFSA membranes are porous to gasses and liquids. If it were impermeable then no water or hydrogen could crossover. 1. Dickinson, E. J. F. & Smith, G. Modelling the proton-conductive membrane in practical polymer electrolyte membrane fuel cell (PEMFC) simulation: A review. Membranes 10, 1–53 (2020).
- Ismail, M. S., Ingham, D. B., Hughes, K. J., Ma, L. & Pourkashanian, M. An efficient mathematical model for air-breathing PEM fuel cells. Energy 135, 490–503 (2014).
Normally in modelling papers it is referred to as porous
- Experimental method and model formulation
Line 117, ‘The anode GDE, the cathode GDE, and the membrane (Nafion 211) were assembled by hot pressing at 130 °C, for 4 min’
Whether Nafion membrane has undergone pretreatment, such as hydrogen peroxide solution and dilute sulfuric acid solution [Solid State Ionics 319 (2018): 110-116]. If yes, there should be a related and detailed description.
Nafion 211 was used as sold; no pre-treatment is required for the membrane. Thicker membranes such as Nafion 117 and 115 usually require treatment to improve their conductivity. This is not the case for Nafion 211 or Nafion 212.
- Results and discussion
- Line 307, ‘ Using the ionic resistance of the cell, the actual water content of the membrane can be determined.’
What is the meaning of ‘the ionic resistance of the cell’? Normally, only the PFSA membrane is ion-conducting, and none of the other components in the fuel cell should be ion-conducting.
Thank you for the comment, the impedance spectroscopy measurement includes the frequency response of the whole cell at 0.6V, we use this voltage to help narrow down the effect of the components of interest however it’s not possible to truly isolate each component therefore impedance will show us the dominating effect at that voltage. That means that we cannot truly say that the high frequency resistance is purely just the membrane component. as the technique is analysing the whole cell. However, you are correct that the ionic components of the cell are the membrane and the ionomer within the catalyst layer. I have changed the text to include membrane and ionomer. (lines 326-330)
- The caption of Table 6 should be ‘Table 6: Increase of lambda between corrected and uncorrected values’. Why use all capital letters suddenly here? It should be avoided. The same applies to Table 8.
I have removed the capitalisation, and water content is now consistently described after section 2 as lambda. After lambda is first defined
- Discussion
Line 373, ‘This work shows that the empirical relationship derived for Nafion 211 for water content vs. water activity is valid up to 120°C with high relative humidity.’
What is the probable reason that after 120 oC the relationship is no longer valid? Is it due to the degradation of membranes at high temperatures (thermal stability issues)? There should be some more details.
Due to the relaxation of the nafion chains there is increased hydrogen crossover which limits the open circuit voltage of the cell. Moreover,considering the data in table 8 the concentration of oxygen at the cathode is 17% at 80oC, 13% at 100oC and 7% at 120oC, and the mol fraction of water is 15, 33 and 66% respectively. Therefore, despite the lack of ‘flooding’ behaviour in the polarisation curve there is still a severe reduction in the available oxygen for reaction, which combined with slow oxygen diffusions and reduction reaction rates in general. Hence, Thefuel cell performance is limited at high temperatures for reasons other than membrane water content, however, increasing the operating pressure to reduce the mol fraction of water in the cell at 100% RH would alleviate this issue.
This has been highlighted in the discussion of table 8 lines 411-423 and the conclusion lines 490-496
Reviewer 2 Report
General comment:
In this manuscript, the authors presented an experimental and numerical study of PEFC's with Nafion® 211 membrane electrolytes at intermediate temperature. This work is interesting. However, the manuscript is not described and organized well. Accordingly, I would not recommend this article to be published on Polymers. Selected comments go as follows.
Comment 1:
In some part of the manuscript, the language the author used is too casual and non-scientific. In addition, there are a few unattended grammar mistakes and typos. The authors should check all of them correct.
Comment 2:
The authors obtained Nyquist plots from EIS measurement in Fig. 3. The corresponding circuit model has to be also mentioned in the manuscript.
Comment 3:
The authors should consistently use lambda or λ.
Comment 4:
There is no unit for temp, RH and current density in table 9. Why did the author particularly choose 0.6V for parameter prediction?
Comment 5:
The result of Fig. 7 shows that modelling cannot fit experimental data very well. The authors should explain why.
Comment 6:
The conclusion should be concisely presented by a synthesis of key points instead of just putting all the results together.
Author Response
I would like to thank the reviewer for all their comments and their time spent reviewing the manuscript, I have highlighted my responses to your comments in red for clarity,
In this manuscript, the authors presented an experimental and numerical study of PEFC's with Nafion® 211 membrane electrolytes at intermediate temperature. This work is interesting. However, the manuscript is not described and organized well. Accordingly, I would not recommend this article to be published on Polymers. Selected comments go as follows.
Comment 1:
In some part of the manuscript, the language the author used is too casual and non-scientific. In addition, there are a few unattended grammar mistakes and typos. The authors should check all of them correct.
Thank you for these comments, I have adjusted the grammar and removed any casual language
Comment 2:
The authors obtained Nyquist plots from EIS measurement in Fig. 3. The corresponding circuit model has to be also mentioned in the manuscript.
Thank you for this comment, several works have already gone into detail about the general shapes of impedance spectra and their equivalent circuits. In this paper we follow work by Pickup’s group; it is a family of publications about the deconvolution of ionic and electronic resistance from impedance data taken where the ohmic resistance is dominant. Since we are following this analysis, it is thought out of scope of this paper to show any other equivalent circuits as we know the one they derived fits. If you wish to see their work, which details many of the types of equivelant circuit, it is referenced in the paper refrerences 51-54. Also for the general shapes mentioned earlier there is a book [Yuan, X. Z., Song, C., Wang, H. & Zhang, J. Electrochemical impedance spectroscopy in PEM fuel cells: Fundamentals and applications. Electrochemical Impedance Spectroscopy in PEM Fuel Cells: Fundamentals and Applications (2010). doi:10.1007/978-1-84882-846-9]
Comment 3:
The authors should consistently use lambda or λ.
The manuscript has been changed to be consistent
Comment 4:
There is no unit for temp, RH and current density in table 9. Why did the author particularly choose 0.6V for parameter prediction?
RH, Temp and Current density have been added. 0.6V was chosen because across all of the experiments, the most accurately modelled values were near 0.6V which makes sense because these values are where the ohmic resistances dominate. The ohmic region is where the membrane performance parameters play the greatest effect.
Also it is discussed in detail that the flooding physics used in the model do not align with the observed experimental behaviour. This makes predicting maximum performance troublesome
Comment 5:
The result of Fig. 7 shows that modelling cannot fit experimental data very well. The authors should explain why.
As explained in comment 4, the model is not perfect and cannot replicate the behaviour observed perfectly. However this is limited to the mass transport region of the polarisation curve, since the ohmic region matched up quite well which is the area of focus for this study. It was demonstrated that the model can be used to predict the membrane behaviour as described lines 390-395
in the discussion section, lines 429 – 435 explain the flooding physics limitations in this paper, however this doesn’t invalidate the ohmic region which is where the model has been used to predict behaviour.
Comment 6:
The conclusion should be concisely presented by a synthesis of key points instead of just putting all the results together.
Thank you for the comment, The conclusion has been made concise.
Reviewer 3 Report
This paper is well written. However some spelling mistakes at the beginning.
Experimental results are in accordance with the proposed model. However it is strange to have a second discussion section in the result and discussion one. Il suggest the authors to better organize the order of this section.
Author Response
Thank you for all your comments, from this point on my comments will be in red for clarity
This paper is well written. However some spelling mistakes at the beginning.
Thank you for your comment! I have adjusted the spelling errors and grammar mistakes
Experimental results are in accordance with the proposed model. However it is strange to have a second discussion section in the result and discussion one. Il suggest the authors to better organize the order of this section.
Thank you for your comments, I have separated the results and discussion sections to make things more clear to read.
Round 2
Reviewer 2 Report
The authors have answered all the comments.